# Machine Learning-Based Regression Framework to Predict Health Insurance Premiums

**DOI:** 10.3390/ijerph19137898

**Published:** 2022-06-28

**Authors:** Keshav Kaushik, Akashdeep Bhardwaj, Ashutosh Dhar Dwivedi, Rajani Singh

**Affiliations:** 1School of Computer Science, University of Petroleum and Energy Studies, Dehradun 248007, India; officialkeshavkaushik@gmail.com (K.K.); bhrdwh@yahoo.com (A.B.); 2Centre for Business Data Analytics, Department of Digitalization, Copenhagen Business School, 2000 Frederiksberg, Denmark; rs.digi@cbs.dk

**Keywords:** artificial intelligence, neural networks, machine learning, health insurance, prediction

## Abstract

Artificial intelligence (AI) and machine learning (ML) in healthcare are approaches to make people’s lives easier by anticipating and diagnosing diseases more swiftly than most medical experts. There is a direct link between the insurer and the policyholder when the distance between an insurance business and the consumer is reduced to zero with the use of technology, especially digital health insurance. In comparison with traditional insurance, AI and machine learning have altered the way insurers create health insurance policies and helped consumers receive services faster. Insurance businesses use ML to provide clients with accurate, quick, and efficient health insurance coverage. This research trained and evaluated an artificial intelligence network-based regression-based model to predict health insurance premiums. The authors predicted the health insurance cost incurred by individuals on the basis of their features. On the basis of various parameters, such as age, gender, body mass index, number of children, smoking habits, and geolocation, an artificial neural network model was trained and evaluated. The experimental results displayed an accuracy of 92.72%, and the authors analyzed the model’s performance using key performance metrics.

## 1. Introduction

We live in a world that is filled with dangers and uncertainties. People, homes, businesses, buildings, and property are all vulnerable to various types of risk, and these risks might differ. These threats include the risk of death, illness, and the loss of property or possessions. People’s lives revolve around their health and happiness. However, because risks cannot always be avoided, the financial sector has devised a number of products to protect individuals and organisations from them by utilizing financial resources to compensate them. As a result, insurance is a policy that reduces or eliminates the expenses of various risks. A policy that protects medical bills is known as health insurance. An individual who has purchased a health insurance policy receives coverage after paying a certain premium. The cost of health insurance is determined by a variety of factors. The cost of a health insurance policy premium varies from person to person since various factors influence the cost of a health insurance plan. Consider age: a young individual is far less likely than an older person to suffer serious health issues. As a result, treating an elderly person is more expensive than treating a young one. As a result, an older individual must pay a higher premium than a younger person. Because [1] numerous factors influence the insurance premium of a health insurance policy, the premium amount varies from person to person.

In healthcare, artificial intelligence is capable of completing many medical-related activities at a much quicker rate in order to forecast or diagnose illnesses/injuries effectively and deliver the best medical therapy to the patient. AI may gather data, process it, and offer the appropriate result to the user. This reduces the time it takes to detect diseases and mistakes, allowing the diagnosis–treatment–recovery cycle to be dramatically shortened. For example, if you choose an online consultation with a doctor, chatbots are used by healthcare professionals or organisations to obtain basic information prior to an appointment with the doctor. This assists the doctor in comprehending the problem before beginning the consultation procedure. As a result, both the doctor and the patient save time.

AI and ML play various roles in the health insurance market, some of which are listed below:The use of chatbots has become an increasingly important aspect of any firm; even healthcare organisations are embracing the technology. Because almost everyone has access to the Internet and a smartphone, interacting with physicians, hospitals, and insurance companies is much easier using chat applications. They are available 24 h a day, seven days a week, making them more effective than human interaction. They employ emotional analysis and natural language processing to better comprehend consumers’ requests and respond to a variety of queries about insurance claims and product choices.Faster Claim Settlements: The time it takes for health insurance claims to be settled is one of the main difficulties for both policyholders and insurers. This might be due to lengthy manual processes or bogus claims. It takes time and effort to manually identify valid claims. However, AI has the potential to significantly lower claim processing times in the future. AI can detect fraudulent claims and learn from previous data to improve efficiency significantly.Personalised Health Insurance Policies: On the basis of an individual’s past data and current health circumstances, insurers can identify and develop a health insurance plan for them. This assists the insurer in providing a proper health insurance plan rather than a health insurance package that clients may or may not utilise efficiently. Customers will also be urged to select a plan that meets their requirements rather than paying for services they may not use.Cost-effectiveness: Insurers are utilising AI to recommend good habits and behaviours to clients, such as exercise and diet, lowering the cost of avoidable healthcare expenditures caused by bad habits.Fraud Detection: Researchers are working on building machines that can evaluate health insurance claims and anticipate fraud. This also aids insurers in resolving legitimate claims more quickly.Faster Underwriting: The health insurance underwriting procedure is lengthy and time-consuming. Fitness trackers, for example, can now collect and analyse vast amounts of data and share it with insurance companies thanks to technological breakthroughs, such as smart wearable technologies. Insurers can find innovative methods to underwrite consumers differently by employing these data. By adopting AI-based predictive analysis, health insurance firms may save time and money.

Even as the healthcare business quickly digitises, enormous amounts of data will inevitably be created and gathered. This will simply increase the workload for healthcare providers since more raw data means more effort. For healthcare professionals and patients, AI can interpret these data and deliver insights based on them. It is a more efficient way to diagnose ailments. Some of the advantages of AI and ML in healthcare are:Clinical Observation-Based Decisions: AI and machine learning can process vast volumes of data in real time and give critical information that can aid in patient diagnosis and treatment recommendations. This translates to improved healthcare services at a reduced cost by evaluating patient data and delivering findings in a couple of minutes. Diabetes or blood sugar devices, for example, may analyse data rather than merely reading raw data and alert you to patterns depending on the information presented, allowing you to take immediate or corrective action.Increased Accessibility: While affluent countries can offer healthcare to the majority of their citizens, underdeveloped countries may struggle. This is owing to a technological gap in healthcare, which results in a drop in the respective country’s health index. Reaching out to individuals in the farthest reaches of the globe is an important task, and the risk of healthcare deprivation is growing. By establishing an efficient healthcare system, AI can assist to alleviate this problem. Digital healthcare will help bridge the gap between poor and wealthy countries by allowing people to better understand their symptoms and obtain treatment as soon as possible.Helps Reveal Early Illness Risks: AI can evaluate enormous amounts of patient medical data and compile it all in one location, which can help reveal early illness risks. It may examine prior and current health issues using the information. Doctors may compare the data and make an accurate diagnosis, allowing them to deliver the best therapy possible. With a large amount of data in one location, AI-powered healthcare applications can assess a wide range of symptoms, diagnose ailments, and potentially forecast future illnesses.Early Detection of Illness: Artificial intelligence can learn from data, such as diagnoses, medical reports, and photographs. This helps detect the beginning of ailments over time as well as implement preventative and mitigation measures.Artificial intelligence also saves time and money by reducing the time and effort required to evaluate and diagnose an ailment. Instead of waiting for a doctor’s consultation to diagnose your sickness, AI will be able to analyse and offer correct inputs to the doctor, allowing the doctor to make the best decision possible and minimising the time it takes to deliver early treatment. People may not need to visit many laboratories for diagnosis if AI can read and evaluate the condition.Expediting Processes: By streamlining visits, interpreting clinical notes, and recording patient notes and treatment plans, AI can assist clinicians in decreasing their administrative load. The benefits of AI in healthcare are numerous since it simplifies operations and offers reliable data in less time.Improve Drug Development: Drug development can take a long time and sometimes miss deadlines for pharmaceutical companies to deliver the proper formula. On the other hand, drug development has never been faster than it is now, thanks to AI. AI allows scientists to concentrate on creating treatments that are both promising and relevant to the needs of patients. It saves time and money when creating medications that might save lives in an emergency.

When it comes to evaluating data, healthcare in India is incredibly complicated and difficult to grasp, and patients often suffer the price. Artificial intelligence (AI) in healthcare can boost efficiency and treatment effectiveness. It can also assist healthcare personnel in spending more time delivering appropriate treatment, lowering burnout among medical experts. Here are a few examples of how AI affect healthcare:In undeveloped or neglected nations, healthcare access is limited.Electronic health records are less burdensome.Antibiotic resistance threats are being reduced.Insurance claims are processed faster.Plans for individual health insurance.

The highlights of this research are:This domain of insurance prediction is not fully explored and requires thorough research. From the proposed machine learning model, patients, hospitals, physicians, and insurance providers could benefit and accomplish their tasks faster and more efficiently.The authors trained an ANN-based regression model to predict health insurance premiums.The model was evaluated against key performance metrics, such as *RMSE*, *MSE*, *MAE*, *r*2, and adjusted *r*2.The overall accuracy of the proposed model was 92.72%.The correlation matrix was plotted to visualise the relationship between various factors with the charges.

This paper is organised as follows: Section 1 starts with the introduction to the topic and concept; this section highlights the latest related work in this domain. Section 3 discusses the working methodology followed in the implementation, and Section 4 shows the results and discussion. Finally, Section 5 contains the conclusion of the entire paper.

## 2. Related Work

The authors identified 245 published research papers from 2017 to the present date from IEEE, ACM, Inderscience, Elsevier, and other highly referenced journals. On the basis of similar research using artificial intelligence and machine learning models to predict the health insurance premium amounts for subscribers, the authors divided and classified these publications with a four-stage selection method. The literature survey for all the selected papers and the categorisation breakdown are illustrated in Table 1.

This helped to shortlist 46 papers that were closely matched and relevant research, as illustrated in Figure 1.

In the healthcare business, predicting health insurance premiums using machine learning (ML) algorithms is still a subject that has to be investigated and improved. The work of [2] presented a computational intelligence technique for estimating healthcare insurance expenditures using a set of machine learning techniques. One essay [3] began by looking at the potential ramifications of using predictive algorithms to calculate insurance prices. Would this jeopardise the idea of risk mutualisation, resulting in new kinds of prejudice and insurance exclusion? In the second stage, the authors looked at how the connection between the company and the insured was altered when the customer realised that the firm had a lot of data about her actual behaviour that was constantly updated.

The goal of the study proposed by van den Broek-Altenburg and Atherly [4] was to find out what customers think about medical insurance by looking at what they talk about on Twitter. The goal was to utilise sentiment classification to find out how people feel about health insurance and doctors. During the 2016–2017 healthcare insurance registration period in the United States, the authors utilised an Application Program Interface (API) to collect tweets on Twitter with the phrases “health insurance” or “health plan”. A policy that decreases or negates the costs of losses caused by different hazards is known as insurance. Several [5] variables impact the cost of insurance. These elements have an impact on the development of insurance plans. Machine learning (ML) can help the insurance industry enhance the efficiency of policy wording.

An article by Nidhi Bhardwaj and Rishabh Anand [6] used individuals’ health data to forecast their insurance premiums. To assess and evaluate the performance of various algorithms, regression was utilised. The dataset was used to train the models, and the results of that training were utilised to make predictions. The model was then tested and verified by comparing the anticipated quantity to the actual data. The accuracy of these models was later compared. Multiple linear regression and gradient boosting algorithms outperformed linear regression and decision trees, according to the findings. Gradient boosting was suitable in this scenario since it required far less computing time to attain the same performance measure, although its performance was equivalent to that of multiple regression.

In the life insurance sector, risk assessment is critical for classifying applicants. Companies utilise screening methodology to produce application decisions and determine the pricing of insurance products. The vetting process may be computerised to speed up applications or programs thanks to the expansion of data and advances in business intelligence. The goal of the study in [7] was to find ways to use predictive analytics to improve risk assessment for life insurance companies. The research was conducted using a real-world dataset with over a hundred characteristics (anonymised). Dimensionality reduction was performed to choose salient features that could increase the models’ prediction potential.

Actuaries utilise a variety of numerical procedures to forecast yearly medical claims expenditure in an insurance business. This sum must be accounted for in the annual financial budgets. Inaccurate estimation usually has a detrimental impact on a company’s overall success. Goundar et al. [8] explained how to build an artificial neural network (ANN) that can predict yearly medical claims. The aim was to lower the mean absolute percentage error by changing factors of the configuration, such as the epoch, learning rate, and neurons, in various layers once the neural network models were constructed. Feed forward and recurrent neural networks were utilised to forecast the yearly claim amounts.

Joseph Ejiyi et al. [9] investigated an insurance dataset from the Zindi Africa competition, which was stated to be from Olusola Insurance Company in Lagos, Nigeria, to demonstrate the efficacy of each of the ML algorithms we employed here. The results showed that, according to a dataset obtained from Zindi, insurance authorities, shareholders, administration, finance professionals, banks, accountants, insurers, and customers all expressed worry about insurance company insolvency. This worry stemmed from a perceived requirement to shield the general public from the repercussions of insurer insolvencies while also lowering management and auditing duties. In this work [10], we offer a strategy for preventing insurance company insolvency. In the past, insolvency prediction approaches, such as multiple regression, logit analysis, recursive partitioning algorithm, and others were applied.

Fauzan and Murfi [11] used XGBoost to solve the issue of claim prediction and evaluate its accuracy. We also compared XGBoost’s performance against that of other ensemble learning methods, such as AdaBoost, Stochastic GB, Random Forest, and Neural Network, as well as online learning methods. In terms of normalised Gini, our simulations suggest that XGBoost outperforms other techniques. People are increasingly investing in such insurance, allowing con artists to defraud them. Insurance fraud is a crime that can be committed by either the customer or the insurance contract’s vendor. Unrealistic claims and post-dated policies, among other things, are examples of client-side insurance fraud. Insurance fraud occurs on the vendor side in the implementation of regulations from non-existent firms and failure to submit premiums, among other things. In this study [12], we compare and contrast several categorisation methods.

Kumar Sharma and Sharma [13] aimed to develop mathematical models for predicting future premiums and validating the findings using regression models. To anticipate policyholders’ choice to lapse life insurance contracts, we employed the random forest approach. Even when factoring in feature interactions, the technique beats the logistic model. Azzone et al. [14] studied how the model works; we employed global and local classification tools. The findings suggest that existing models, such as the logistic regression model, are unable to account for the variety of financial decisions.

Understanding [15] the elements that influence a user’s health insurance premium is critical for insurance firms to generate proper charges. Premium should always be a user’s first concern when making suitable selections. The majority of characteristics that contribute to the cost of health care premiums are BMI, smoking status, age, and kids, according to the output, which revealed that these four parameters have a strong correlating effect on health insurance rates.

Premiums are determined by health insurance companies’ private statistical procedures and complicated models, which are kept concealed from the public. The goal of this study [16] is to see if machine learning algorithms can be used to anticipate the pricing of yearly health insurance premiums on the basis of contract parameters and business characteristics. The goal of this article [17] is to use a strong machine learning model to estimate the future medical costs of patients on the basis of specific parameters. Using the simulation results, the elements that influence individuals’ medical expenditures were determined.

The Japanese government has mandated that insurers develop a population health management strategy. To assess the strategy [18], a cost estimate is required. A standard linear model is not suited for the prediction since one insured patient might have several conditions. Using a quantitative machine learning technique, we created a medical cost forecasting model. The historical uniformity of health care expenses in a major state Medicaid programme was investigated in this research. The expenses were forecasted using predictive machine learning algorithms, particularly for high-cost, high-need (HCHN) patients. The findings of Yang et al. [19] indicated a high temporal link and showed potential for utilising machine learning to forecast future health care spending. HCHN patients had a stronger temporal association, and their expenditures may be anticipated more accurately. Including additional historical eras improved forecasting accuracy.

Some individuals who are economically disadvantaged will be unable to cover treatment-related fees.

According to our behaviour and genetics, the necessity for health insurance varies as we grow older. Health insurance is becoming increasingly important as people’s lifestyles and ailments change. Because a medical problem can strike anybody at any moment and have such a significant psychological and economic impact on the individual, it is difficult to predict when one will occur. With this background in mind, this research [20] aimed to forecast the cost of health insurance using the following contributions: age, gender, region, smoking, BMI, and children.

The K-means algorithm [21] and the Elbow technique were used in this study to properly arrange people into an appropriate number of clusters on the basis of similarities. On the basis of this analysis, the health insurance premium quotation was predicted for each group of people using the specified criteria. Predicting the cost of people’s health insurance is a valuable way to increase healthcare accountability. In order to forecast insurance premiums for people in this research [22], Sailaja et al. employed several regression models to assess personal health information. A lot of things impact the cost of insurance rates. The use of a Stacking Regression model to anticipate insurance prices for people might help health insurers. Dutta et al. [23] estimated the cost of health insurance that the patient was responsible for paying. To accomplish the best prediction analysis, several data mining regression methods, such as decision tree, random forest, polynomial regression, and linear regression, were used. A study of the actual and expected expenditures of the prediction premium was made, and a graph was displayed as a result, allowing us to select the best-suited regression technique for insurance policy forecasting.

## 3. Research Methodology

In this paper, the authors used the Python programming language for the implementation and trained the machine learning-based model for the prediction of health insurance premiums. Initially, the dataset and the necessary python libraries and packages were imported. The dataset consisted of over 1300 entries and seven columns, namely charges, smoking, region, children, BMI, sex, and age. This dataset was used to predict the health insurance premium. Thereafter, an exploratory data analysis was performed. In this step, the dataset was checked for null values. Since there were no null values in the dataset, the statistical summary of the dataset was analysed. The statistical summary included the count, mean, standard deviation, and various other statistics related to the columns available in the dataset—age, BMI, number of children, and health insurance charges. The dataset link is given at the end of the paper in the Data Availability Statement. The entire methodology followed in this paper is shown in Figure 2.

### 3.1. Step 1: Performing the Data Analysis and Feature Engineering

In this step, the dataset was analysed to check the relationship between the various columns. As shown in Table 2, it was observed that the southeast region had the highest charges and body mass index. The dataset was grouped by age, and then the relationship between age and charges was analysed.

In this step, the unique values in the sex, smoking, and region columns were checked, and the categorical variables were converted to numerical variables.

### 3.2. Step 2: Data Visualisation

In the previous step, the dataset was cleaned so that the model could be trained and visualised. In this step, the data was visualised to obtain useful information. In Figure 3, the histogram is plotted for all the columns present in the dataset for a visual glimpse.

After that, the pairplot diagram was plotted, as illustrated in Figure 4. Pairplot diagrams are used to figure out which attributes best explain the connection between two variables or form the most separated groups. Drawing basic lines or making a linear distinction in our dataset also aided in the formation of some simple categorisation models.

The pairplot diagram showed the relationship between the various columns present in the dataset. A pairplot is a grid that shows all the different scatter plots with all the different combinations in our data. After plotting the pairplot diagram, the regplot was plotted, as shown in Figure 5. We can see that as age increased, charges tended to increase as well. Therefore, there is a linear relationship between the charges and age.

Regplot is a programme that plots data and fits a linear regression model. To evaluate the regression model, there are several mutually incompatible alternatives. In Figure 6, there is a straight line that passes through the data, and it seems that body mass index (BMI) tends to increase a little bit. It is possible that the charges also tend to increase slightly.

### 3.3. Step 3: Training and Evaluating a Linear Regression Model

In this step, the authors trained the linear regression model, but before training the model, the dataset was cleaned. Only the numerical values were taken, and the data were scaled. A standard scaler was used to scale the data. Scaling the data is important before feeding the data to the model. Once the data was scaled completely, the linear regression model was trained. The accuracy of the linear regression model came out to be 75.09%. After that, the linear regression model was evaluated by finding the Root Mean Square Error (*RMSE*), Mean Squared Error (*MSE*), Mean Absolute Error (*MAE*), and adjusted *r*2 score. The formulas used for the calculation of all the parameters mentioned are given below.
RMSE= float(format(np.sqrt(mean_squared_error(y_test_orig, y_predict_orig)),’.3f’))
MSE= mean_squared_error(y_test_orig, y_predict_orig)
MAE= mean_absolute_error(y_test_orig, y_predict_orig)
r2=r2_score(y_test_orig, y_predict_orig)
adj_r2=1−(1−r2)×(n−1)/(n−k−1)

The output of the evaluation is shown in Table 3.

## 4. Results and Discussion

The final step i.e., training and evaluating an ANN-based regression model is discussed in this section. Initially, the entire dataset is split into 20% testing data and 80% training data. In training the ANN model, the authors have used keras sequential model in which five dense layers are added and ‘relu’ activation function is used. Adam optimiser is used to optimise the performance of the model. Table 4 shows the model summary. The total trainable parameters are 38,351 whereas there are 0 non-trainable parameters.

The model was trained for 100 epochs, and the batch size was 20 with a validation split equal to 0.2. The accuracy of this model came out to be 92.72%, and the validation and training loss is plotted in Figure 7.

Moreover, the model predictions and true values were also plotted to see the relationship between them. Figure 8 shows the plot of model predictions vs. true values, whereas Figure 9 shows the inverse transform plot of model predictions vs. true values.

Once the ANN model was trained and the accuracy was calculated, then the performance of the model was evaluated using the same performance metrics, i.e., *RMSE*, *MSE*, *MAE*, *r*2, and adjusted *r*2. Table 5 shows the comparison between the evaluation metrics of our trained ANN model and the linear regression model. From the comparison, it is clear that our trained model had better performance.

Here, one can compare Table 4 and can conclude that the evaluation metrics of our trained model are better than those of the linear regression model. Finally, the correlation matrix was plotted to see the positive and negative relationships among the multiple factors. Here, after observing the correlation matrix in Figure 10, we can conclude that charges are positively related to smoking and age, whereas southwest and northwest regions are negatively related to charges.

## 5. Conclusions

In the field of health insurance, machine learning is well-suited to tasks that are often performed by people at a slower speed. AI and machine learning are capable of analysing and evaluating large volumes of data in order to streamline and simplify health insurance operations. The impact of machine learning on health insurance will save time and money for both policyholders and insurers. AI will handle repetitive activities, allowing insurance experts to focus on processes that will improve the policyholder’s experience. Patients, hospitals, physicians, and insurance providers will benefit from ML’s ability to accomplish jobs that are currently performed by people but are much faster and less expensive when performed by ML. When it comes to exploiting historical data, machine learning is one component of cognitive computing that may address various challenges in a broad array of applications and systems. Forecasting health insurance premiums is still a topic that has to be researched and addressed in the healthcare business. In this study, the authors trained an ANN-based regression model to predict health insurance premiums. The model was then evaluated using key performance metrics, i.e., *RMSE*, *MSE*, *MAE*, *r*2, and adjusted *r*2. The accuracy of our model was 92.72%. Moreover, the correlation matrix was also plotted to see the relationship between various factors with the charges. This domain of insurance prediction has not been fully explored and requires thorough research.

## Figures and Tables

**Figure 1 ijerph-19-07898-f001:**
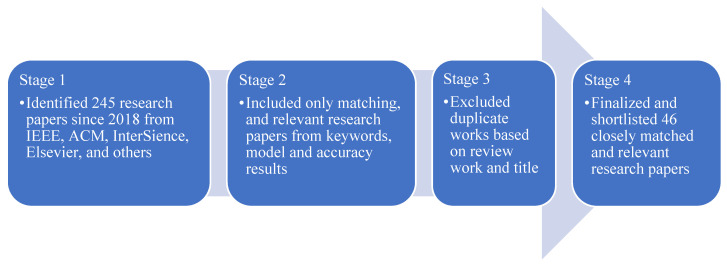
Research papers selection process.

**Figure 2 ijerph-19-07898-f002:**
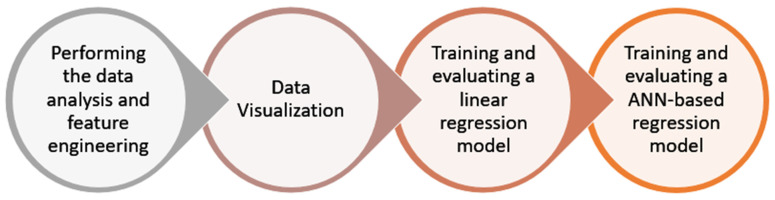
Machine learning-based regression framework.

**Figure 3 ijerph-19-07898-f003:**
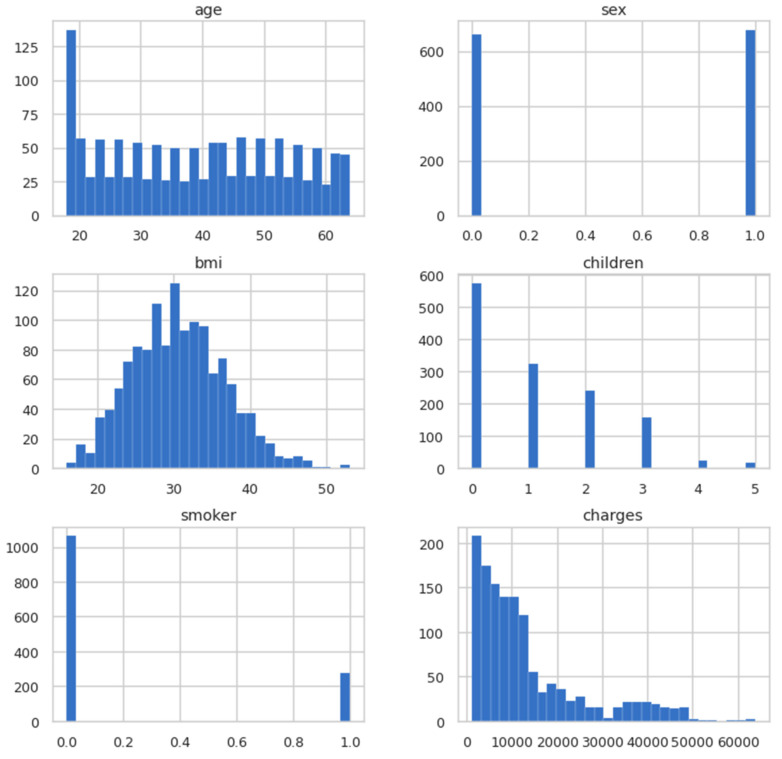
Histogram plots for columns.

**Figure 4 ijerph-19-07898-f004:**
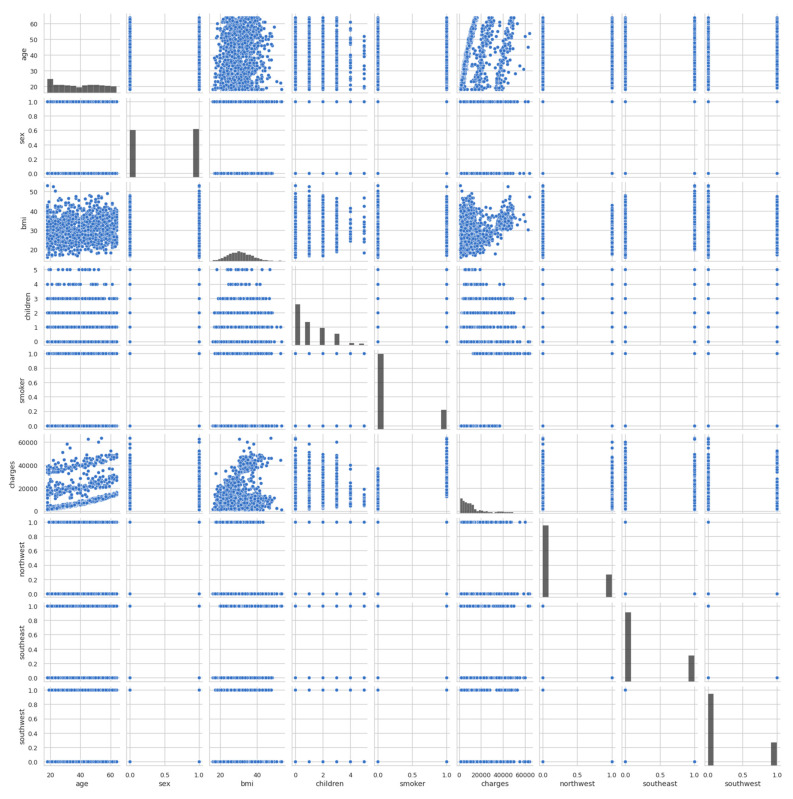
Pairplot diagram of entire dataset.

**Figure 5 ijerph-19-07898-f005:**
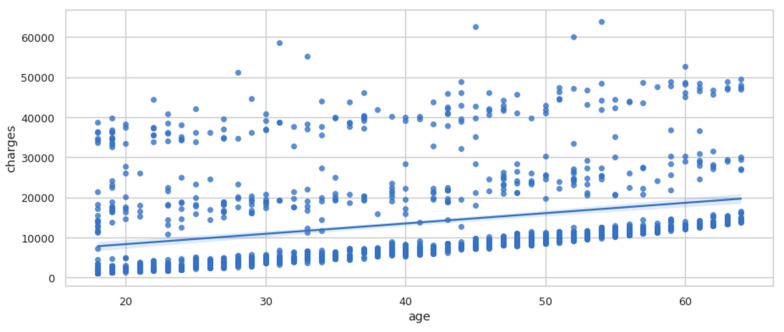
Regplot of charges vs. age.

**Figure 6 ijerph-19-07898-f006:**
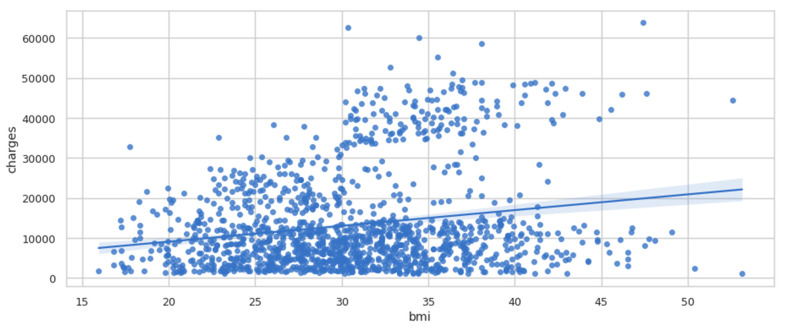
Regplot of charges vs. BMI.

**Figure 7 ijerph-19-07898-f007:**
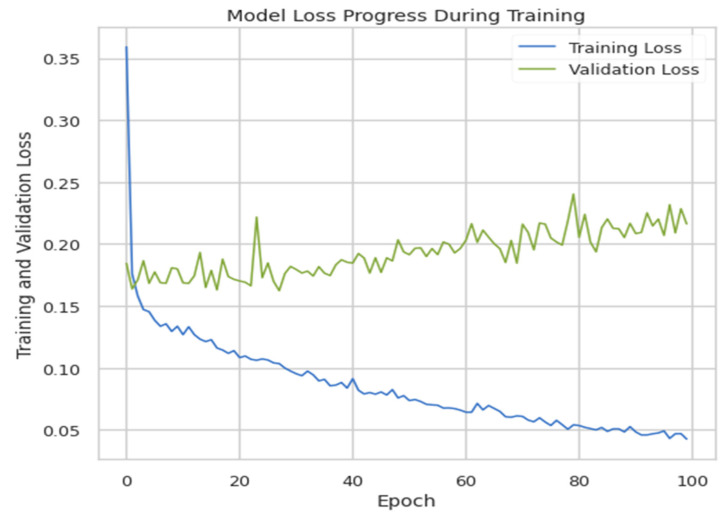
Training loss vs. validation loss.

**Figure 8 ijerph-19-07898-f008:**
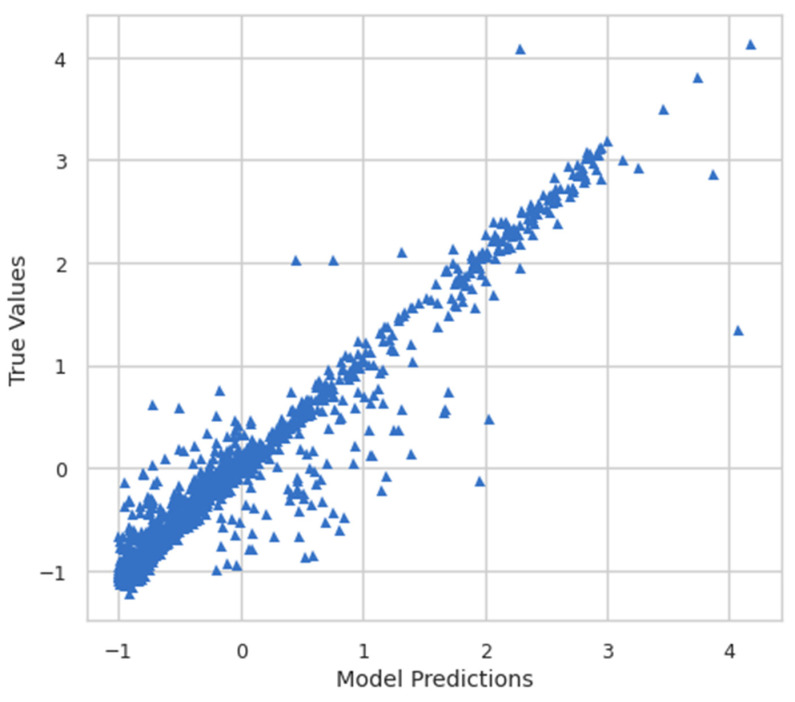
Model predictions vs. true values.

**Figure 9 ijerph-19-07898-f009:**
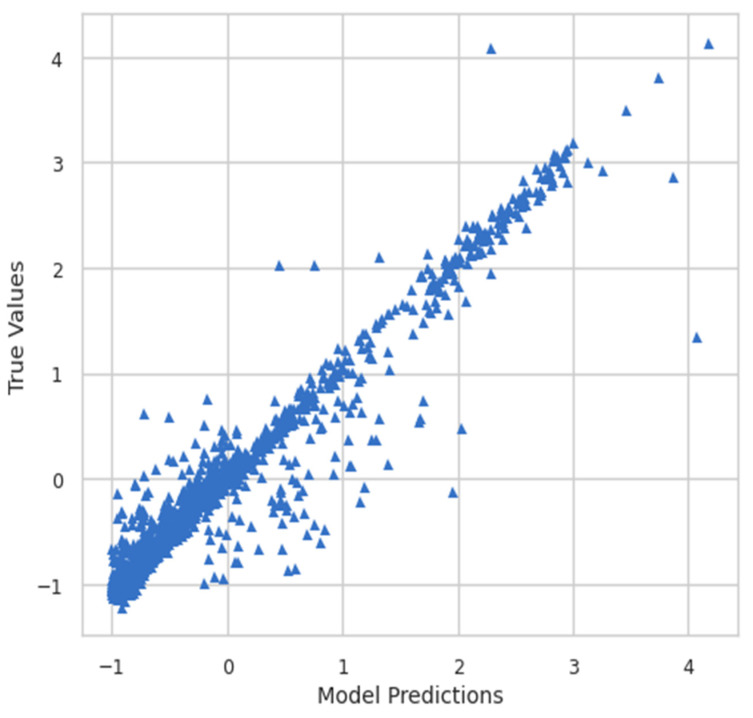
Inverse transform of model predictions vs. true values.

**Figure 10 ijerph-19-07898-f010:**
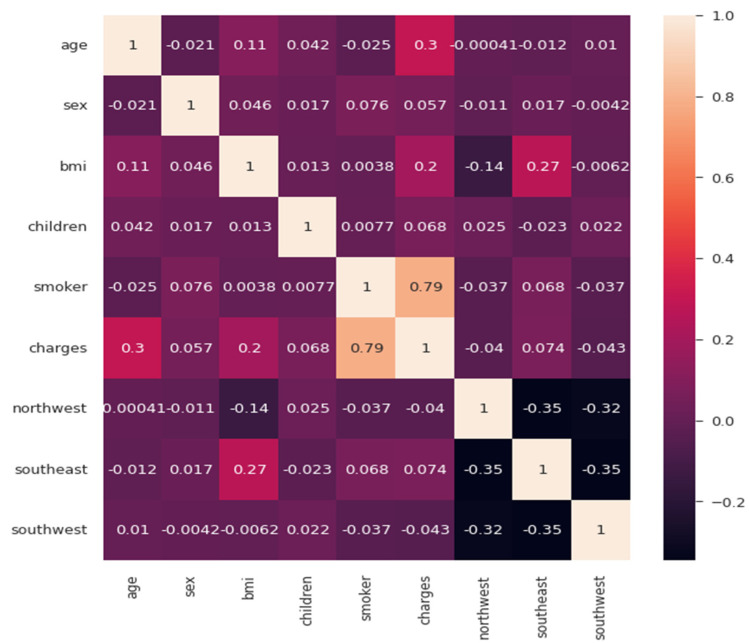
Correlation matrix.

**Table 1 ijerph-19-07898-t001:** Research literature classification.

Literature Classification	Stage 1	Stage 2	Stage 3	Stage 4	Breakdown
Health insurance prediction	54	38	23	10	22.04%
Premium calculation	53	37	22	10	21.63%
Machine learning	77	54	32	15	31.43%
Artificial intelligence	61	43	26	12	24.90%
Neural networks	245	172	103	46	

**Table 2 ijerph-19-07898-t002:** Relationship between the region and charges.

Region	Age	BMI	Children	Charges
Northeast	39.268519	29.173503	1.046296	13,406.384516
Northwest	39.196923	29.199785	1.147692	12,417.575374
Southeast	38.939560	33.355989	1.049451	14,735.411438
Southwest	39.455385	30.596615	1.141538	12,346.937377

**Table 3 ijerph-19-07898-t003:** Evaluation metrics for the linear regression model.

Evaluation Metrics	Value
*RMSE*	0.499
*MSE*	0.24908696
*MAE*	0.3445451
*r*2	0.7509130368819994
adjusted *r*2	0.7494136420701529

**Table 4 ijerph-19-07898-t004:** ANN model summary.

Layer (Type)	Output Shape	Number of Parameters
Dense (dense)	(None, 50)	450
activation (activation)	(None, 50)	0
dense_1	(None, 150)	7650
activation_1 (activation)	(None, 150)	0
dense_2 (dense)	(None, 150)	22,650
activation_2 (activation)	(None, 50)	0
dense_3 (dense)	(None, 50)	7550
activation_3 (activation)	(None, 50)	0
dense_4 (dense)	(None, 1)	51

**Table 5 ijerph-19-07898-t005:** Comparison of the evaluation metrics for the trained ANN model vs. linear regression model.

Evaluation Metrics	ANN Value	Linear Value
*RMSE*	0.27	0.499
*MSE*	0.07275635	0.24908696
*MAE*	0.1432731	0.3445451
*r*2	0.9272436488919791	0.7509130368819994
adjusted *r*2	0.9268056874105162	0.7494136420701529

## Data Availability

The dataset used in this research is publicly available at https://www.kaggle.com/datasets/noordeen/insurance-premium-prediction (accessed on 20 June 2022).

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
