# Peer review of "Machine Learning-Based Regression Framework to Predict Health Insurance Premiums"

_ijerph, 2022, doi:10.3390/ijerph19137898_

Round 1

Reviewer 1 Report

Although the subject of the manuscript is of interest to the journal, I have several observations the authors may want to consider:

No clear takeaways of the study are highlighted in the abstract. Moreover, the conclusions do not pinpoint what are the benefits and implications of this study. What is the main conclusion of the study? 

Although it is honorable for the authors to attempt to summarize the uses of AI in health services, the introduction loses focus and makes no reference to the scientific literature. Also, the need and usefulness of the endeavor are not highlighted. How does this analysis differ for previous studies?

Data sources, information about the sample size, descriptive statistics are not provided.

The article should be thoroughly proof-read by a native speaker because the poor language leads to misunderstandings. 

Author Response

Although the subject of the manuscript is of interest to the journal, I have several observations the authors may want to consider:

No clear takeaways of the study are highlighted in the abstract. Moreover, the conclusions do not pinpoint what are the benefits and implications of this study. What is the main conclusion of the study?

Response: Many thanks for the comment. As per your suggestion, the takeaways have been highlighted in the abstract and conclusion of the paper.

Although it is honorable for the authors to attempt to summarize the uses of AI in health services, the introduction loses focus and makes no reference to the scientific literature. Also, the need and usefulness of the endeavor are not highlighted. How does this analysis differ for previous studies?

Response: The highlights of this research that makes this paper different from the previous studies has been highlighted in the introduction section. Many thanks for your valuable suggestions.

Data sources, information about the sample size, descriptive statistics are not provided.

Response: The information about the dataset used in this paper is added and highlighted in the section 3 – research methodology.

The article should be thoroughly proof-read by a native speaker because the poor language leads to misunderstandings.

Response: The article is now thoroughly proofread for the grammatical mistakes. Many thanks.

Reviewer 2 Report

Research subject and problem

The research problem has been correctly defined and the topic (title) of the article corresponds well to it. The research problem is in the scope of interest of the IJERPH journal. In the title of the article, it is possible to add in brackets " on the example of population data from XYZ country/region"

Objectives and tasks

The research purpose has been well correlated with the research subject matter. The research tasks are reduced to the steps of model building. These could be further distinguished in the introduction.

Research gaps

The research gaps have been briefly described in general terms. It is advisable to clearly arrange them into theoretical, methodological and empirical gaps. This is important because research gaps are the basis for the formulation of the research problem. It is advisable to add in the introduction and the literature review (“Related Works”).

Questions and hypotheses

Research questions and hypotheses were not formulated. The reason is the methodological nature of the article and the research.

Compliance of the content with the main objective (title)

The content of the article fully corresponds to the aim of the research and the formulated title.

Scientific value and originality

The scientific value of the research is high. It was conducted reliably. The authors' good knowledge of the research technique was demonstrated. The article is an original scientific work.

Methodology (selection of methods and tools)

The authors decided to choose the linear regression method. A critical review of available methods should be added and the reasons for the choice of the method that was used should be clearly indicated. The literature review indicates the possible use of other methods. It is advisable to create a scheme of conducted research (research framework).

Data and research test

The resulting characteristics of the database are presented. However, the description of basic information about the creation and size of this database is missing. This concerns the number of observations (~1500?), the time of observations (period, years) and their location (country/region).

Interpretation of the results

The section "Results and Discussion" contains the last step of model estimation - its training and evaluation. It is therefore not the part of the article corresponding to the title of this section. While "Results" can be filled with the results stage of training and evaluation of the model, "Discussion" must be created.

Discussion and conclusions

As indicated above, the "Discusion" section must be created and it is intended to contain a polemic with the results of other studies, estimations of other models. Such a point is a logical continuation of the literature review (Related Works), precisely as a critical evaluation of the effects of one's own achievement against other similar ones. In the "Discussion" section the limitations of the conducted research must be demonstrated and evaluated.

The importance of solving the problem for theory and practice

The article represents an important voice in research in the area of machine learning and its use in insurance. Its positive application value is evident. What is missing, however, is a model formula for the estimated model so that it can be used as a relationship between the explained variable and the explanatory variables. The model formula should be included in the methodology section.

Structure and composition

The structure of the article needs to be modified and expanded. The title "Research Methodology" should be changed to "Materials and Methods" and the content expanded to include the characteristics of the database indicated above as expected. The item "Results and Discussion" should be split into two "Results" and "Discussion". This second one should be rewritten. The "Conclusions" item has too sketchy content. They should be focused on the general results of the research and indicate directions for further research. The bibliography of the article needs to be enriched. The 23 entries are far too few because even the results of the literature review indicate an initial set of 245 and a final set of 46. The changes in structure made should be guided by the "IJERPH Instructions for authors".

Formal requirements (language, edition)

The article needs standard linguistic and editorial correction. The form of defining the person of the author should be standardized - it is not one “author” but multiple “authors”.

Author Response

Many thanks for the comment. We have updated and highlighted the changes in the paper as per suggestion.

Reviewer 3 Report

The topic of the paper is interesting. However, the authors must revise the paper to address main problems:

1. Where did you get the data?

2. In section 2, you shouldn't write such as: "In this paper [11]...." and shouldn't use "we" to avoid misunderstanding that the papers are your work.

3. How did you calculate the accuracy of 92.72%?

4. Did you check the 4 main assumptions of linear regression model before fitting the data?

5. How big is your data? How did you split data to the training and testing sets? 

6. What is the contribution of your research compare to the 245 published research papers that you mentioned in Section 2?

Author Response

. Where did you get the data?

Response: Many thanks for the comment. The information about the dataset is added and highlighted in the research methodology section.

  1. In section 2, you shouldn't write such as: "In this paper [11]...." and shouldn't use "we" to avoid misunderstanding that the papers are your work.

Response: Many thanks for pointing out to this mistake. The same has been rectified and highlighted under section 2.

  1. How did you calculate the accuracy of 92.72%?

Response: The accuracy is calculated by the ANN model that we have used as mentioned in table 3. The same has been highlighted in the paper for your reference.

  1. Did you check the 4 main assumptions of linear regression model before fitting the data?

Response: Yes.

  1. How big is your data? How did you split data to the training and testing sets?

Response: The dataset has more than 1300 entries, the same has been highlighted in the paper. The dataset is splitted into two parts 20% testing and 80% training.

  1. What is the contribution of your research compare to the 245 published research papers that you mentioned in Section 2?

Response: Many thanks for asking this question. For your reference, the highlights of the research done are highlighted in the introduction section.

Reviewer 4 Report

This is an interesting and useful study. However, this study needs some revisions. Moreover, the following queries need to be clarified in this study.

1.         What are the limitations of this model? What do these limitations affect the results or conclusions?

2.         The methods need to be clearly written and explained so that all readers could recreate the study. Moreover, I suggest that authors could list the algorithm of the model in the appendix / supplementary material to let readers realize the detailed contents.

Author Response

  1. What are the limitations of this model? What do these limitations affect the results or conclusions?

Response: Our model is sensitive to outliers and is prone to overfitting. But, these limitations do not affect the accuracy of the model much and therefore we managed to achieve good accuracy.

  1. The methods need to be clearly written and explained so that all readers could recreate the study. Moreover, I suggest that authors could list the algorithm of the model in the appendix / supplementary material to let readers realize the detailed contents.

Response: Thanks for your valuable suggestion. As per your suggestion, the methodology has been improved and highlighted. Many thanks.

Round 2

Reviewer 4 Report

The paper is much improved and worthy of publication.